# Pyrolysis of Porous Organic Polymers under a Chlorine Atmosphere to Produce Heteroatom-Doped Microporous Carbons

**DOI:** 10.3390/molecules26123656

**Published:** 2021-06-15

**Authors:** Wojciech Kiciński, Sławomir Dyjak, Mateusz Gratzke

**Affiliations:** Institute of Chemistry, Military University of Technology, 2 Kaliskiego Street, PL 00-908 Warsaw, Poland; slawomir.dyjak@wat.edu.pl (S.D.); mateusz.gratzke@wat.edu.pl (M.G.)

**Keywords:** organic polymer carbonization, chlorine, heteroatom-doped carbon, microporosity, ammonia annealing

## Abstract

Three types of cross-linked porous organic polymers (either oxygen-, nitrogen-, or sulfur-doped) were carbonized under a chlorine atmosphere to obtain chars in the form of microporous heteroatom-doped carbons. The studied organic polymers constitute thermosetting resins obtained via sol-gel polycondensation of resorcinol and five-membered heterocyclic aldehydes (either furan, pyrrole, or thiophene). Carbonization under highly oxidative chlorine (concentrated and diluted Cl_2_ atmosphere) was compared with pyrolysis under an inert helium atmosphere. All pyrolyzed samples were additionally annealed under NH_3_. The influence of pyrolysis and additional annealing conditions on the carbon materials’ porosity and chemical composition was elucidated.

## 1. Introduction

For decades, carbonaceous materials have remained at the frontier of advanced purification/separation process technologies, renewable energy management, and catalysis. This is true not only for advanced carbon types (e.g., graphene or nanotubes) but also for traditional forms, such as activated carbons, graphite, and carbon blacks [1,2,3,4]. With increasing political and social pressure to develop a more sustainable economy, traditional carbon-based materials reappeared as an indispensable ingredient of renewable energy technologies and the so-called green chemical syntheses [5]. More recently, carbonaceous materials dominated the area of metal-free and single-atom catalysis [6,7,8]. This is because, unlike any other known material, carbon exhibits a set of seemingly self-excluding characteristics, such as good thermal and electrical conductivity, chemical inertness (resistivity in gas and liquid corrosive media), high specific surface area and extensive porosity, low density, good mechanical strength, and virtually unlimited abundance in the environment. Although in theory, carbon-based materials appear to be wonder materials, in reality, their advanced applications are not straightforward. This is because carbonaceous materials constitute a very diversified family of solid-state substances with extremely varied properties [9]. Even an individual class of carbon materials, such as microporous carbons (which are especially interesting for purification, energy storage, and catalysis), constitutes an extremely varied group of materials. For this reason, new and reproducible synthesis methods for advanced porous carbons are under scrutiny [10,11]. Pyrolytic decomposition of organic matter and further high-temperature annealing remains the method of choice to produce functional carbon-based materials.

Biomass and fossil fuels are the most common feedstock for carbon-based materials. Nevertheless, some advanced applications require carbons of high purity and homogeneity. In this regard, synthetic polymers became an attractive feedstock for the controllable and replicable synthesis of new types of porous carbons [12,13]. Among many advantages (such as high purity and predetermined porosity of porous polymers), synthetic polymers allow precise heteroatom doping [14]. However, while the feedstock is paramount and countless variations have already been proposed, the conditions of its transformation into carbon, i.e., pyrolysis, are somewhat less explored. Pyrolysis used to obtain carbonaceous materials (referred to as carbonization) is usually performed under an oxygen-deficient or inert atmosphere at temperatures in the range of 500–1000 °C. In laboratory studies, nitrogen and argon are routinely used as protective pyrolysis atmospheres. However, carbon is also known to be stable under high-temperature annealing in reactive atmospheres, such as chlorine, bromine, and iodine atmospheres [15]. In fact, chlorine thermochemical extraction of metal carbides is a well-scrutinized approach to create microporous carbon materials with precise control of micropore size distributions [16,17]. Chlorination under high temperatures is also a longstanding and well-established method for removing metallic impurities from graphite and other carbonaceous materials [9,18]. Nevertheless, pyrolysis of synthetic polymers under a Cl_2_ atmosphere is poorly investigated. Only a few reports have explored carbon materials synthesis via chlorination of organometallic compounds [19,20].

Herein, we studied the pyrolysis of heteroatom-containing porous synthetic polymers (cross-linked thermosetting resins) under a chlorine atmosphere of two different Cl_2_ concentrations and compared the results to those obtained through pyrolysis under an inert helium atmosphere. Moreover, all the pyrolyzed polymers were additionally subjected to ammonia annealing. Heteroatom-containing porous organic polymers obtained via sol-gel synthesis were utilized as model materials to perform these analyses. We consider these polymers appropriate representative raw materials since they were characterized in detail (concerning their syntheses, structure, composition, morphology, texture, etc.) in our previous reports [21]. For all the details concerning the utilized polymer materials, please refer to reference [21]. The aim of this study was to explore new approaches for the synthesis of heteroatom-doped microporous carbons with well-defined and reproducible characteristics. The investigation of the transformation of organic polymers into chars under Cl_2_ is of particular interest since high-temperature chlorine annealing does not destroy carbonaceous materials, yet chlorine itself exhibits highly oxidizing properties and hence might positively influence the thermal decomposition of organic feedstock into microporous carbonaceous materials [15,16,17,18,19,20].

## 2. Experimental Section

### 2.1. Synthesis of Porous Organic Polymers

Three types of porous organic polymers were prepared via a sol-gel polycondensation reaction and hence are referred to as gels. The syntheses were performed according to a procedure presented elsewhere [21]. The obtained polymers differed only by the heteroatom present in the heterocyclic aldehyde used for synthesis. The organic gel obtained from 2-formylpyrrole is referred to as the N-doped polymer, the organic gel obtained from 2-formylthiophene is referred to as the S-doped polymer, and the organic gel obtained from 2-furaldehyde is referred to as the O-doped polymer (Figure 1). In brief, 4.0 g of resorcinol and a corresponding amount of heterocyclic aldehyde (either 2-formylpyrrole, 2-formylthiophene, or 2-furaldehyde) was dissolved in 100 mL of methanol or a methanol/water mixture. The molar ratio of resorcinol to aldehyde was 0.5 in each case. Excess of an aldehyde with respect to resorcinol yields extensive cross-linkage and hence thermosetting polymeric structure. The mixture was stirred at room temperature, acidified with 2.0 mL of concentrated hydrochloric acid, closed in a tight container, and placed in an oven set at 60 °C for curing. The transition from a liquid sol to a solid gel took approximately 7 h for 2-formylthiophene, 1 h for 2-furaldehyde, and only 5 min for 2-formylpyrrole. After polycondensation (curing at 60 °C for 24 h) and drying to a constant mass, approximately 10.0–11.0 g of organic polymers was obtained in each case. The syntheses and carbonization scheme are presented in Figure 1.

### 2.2. Pyrolysis under Chlorine (or Helium) and Additional Ammonia Annealing

Carbonization of the studied polymers was performed in a vertical split furnace with a quartz tubular reactor equipped with a ceramic porous shelf (a gas-permeable partition) where the examined samples were placed. The utilized gases were introduced into the bottom of the reactor and flowed steadily upwards, percolating through the entire volume of the powdered polymer material. This design ensures excellent contact between the flowing gas and sample particles due to the fluidizing type of mixing. Both ends of the quartz reactor were equipped with stainless steel gas-tight connections with FKM o-rings, providing a leak-free and oxygen-free atmosphere. Helium (BiP+ purity, Air Products, Warsaw, Poland) and chlorine (99.999% purity, Linde Gas, Cracow, Poland) flows were transferred by stainless steel pipes and managed by thermal mass flow controllers (Bronkhorst High-Tech, Ruurlo, The Netherlands). Prior to carbonization, the reactor with the previously inserted sample was thoroughly purged with helium (for 120 min at a flow rate of 100 mL min^−1^) to remove air and moisture. Afterward, the polymer powder was gradually heated at a heating rate of 5 °C min^−1^ up to 800 °C and kept at this temperature for 60 min. Two different Cl_2_ concentrations were used: either 10% (10 mL min^−1^ Cl_2_ + 90 mL min^−1^ He) or 90% (90 mL min^−1^ Cl_2_ + 10 mL min^−1^ He). During the whole carbonization experiment, the total gas flow rates were kept at 100 mL min^−1^. The chlorine excess and the pyrolysis byproducts were trapped in a liquid nitrogen cold trap. Once the pyrolysis was completed, the Cl_2_ flow was stopped, and the sample was maintained at the same temperature (800 °C) under a flow of He (100 mL min^−1^) for the next hour to perform the preliminary purification of the resultant carbonaceous materials (removal of the adsorbed chlorine) and to rinse the internal volume of the quartz reactor. Upon cooling, the obtained sample was weighed, and half of the sample mass was subjected to further annealing under a flow (100 mL min^−1^) of NH_3_ (99.9999% purity, Linde Gas) at 800 °C for 60 min. For comparison, the studied polymers were also pyrolyzed under a helium atmosphere (under the same conditions used in chlorine pyrolysis) and then the obtained chars were also additionally annealed under ammonia.

### 2.3. Characterization

N_2_ adsorption–desorption measurements were performed at liquid nitrogen temperature using an Autosorb IQ analyzer (Quantachrome Instruments (a brand of Anton Paar), Boynton Beach, FL, USA). The samples were outgassed at 300 °C for 10 h prior to analysis. The Brunauer–Emmett–Teller specific surface area (S_BET_) was calculated from the N_2_ adsorption isotherm, taking into consideration the IUPAC recommendations [22]. The total pore volume (V_0.99_) was estimated from the volume adsorbed at a relative pressure (pp_0_^−1^) of ~0.99. The volume of micropores (V_mi_) was estimated using the Dubinin–Radushkevich (DR) model. When analyzing the results obtained with the cryogenic N_2_ physisorption (S_BET_ and pore volume values), one must always consider the limitations of the accuracy of this method [22,23]. An elemental analysis to assess carbon, hydrogen, nitrogen, and sulfur wt.% content was performed using a Vario EL cube apparatus (Elementar, Germany). The oxygen content (O%) was calculated as O% = 100% − C% − H% − N% − S%.

## 3. Results and Discussion

The undertaken research had two well-focused aims: (1) to study the influence of carbonization under chlorine on the porosity and elemental composition of porous carbonaceous materials derived from heteroatom-containing synthetic polymers and (2) to elucidate how pyrolysis under oxidative Cl_2_ compares with pyrolysis under an inert gas. Some chlorine chemisorbs on the carbon surface upon high-temperature chlorine treatment [15,16]. However, it can be easily removed by annealing under hydrogen or ammonia. Consequently, the carbonaceous materials obtained upon Cl_2_ pyrolysis were further treated with NH_3_. Such treatment not only removes chlorine residues (6 ‒Cl_(chemisorbed)_ + 2NH_3_ → 6HCl + N_2_) but also activates carbon materials via carbon gasification (3C + 4NH_3_ → 3CH_4_ + 2N_2_). Consequently, annealing under ammonia has multiple advantages as it removes chemisorbed chlorine and some portion of the most unstable carbon atoms (disordered/defective phase prone to gasification). Additionally, NH_3_ annealing introduces N-functional groups to the surface of porous carbons, but overall, visible mass loss is expected due to the intense methanation of elemental carbon.

Table 1 presents the elemental composition (wt.%) of the three studied organic polymer types. The S- and N-doped polymers contain up to 20 and 10 wt.% of the corresponding heteroatoms. The S-doped polymer exhibits the lowest relative content of carbon, hydrogen, and oxygen due to the high atomic mass of sulfur. All polymers contain oxygen since they are derived from resorcinol; however, as expected, the furfural-based resin shows the highest O content.

Carbonization of cross-linked thermosetting resins is a complex process, even more so if such resins contain additional heteroatoms. They yield hard and poorly crystalized carbons resistant to graphitization. Since pyrolysis (and destructive distillation) of organic matter is one of the most important processes in the production of carbonaceous materials, it is not surprising that thermal decomposition, carbonization, coking, and further graphitization of heteroatom-containing polymers are studied extensively. However, the carbonization/coking mechanisms of organic and carbonaceous materials with a variety of coexisting defects, including O, N, and S heteroatoms, pentagons, and heptagons, are still not well understood [24]. O, N, and S heteroatoms affect these processes significantly but in very different manners. Experimental data show that during carbonization, most of the initial oxygen in the raw materials is lost at the beginning of pyrolysis (lower temperature region), and hence the intermediate species formed after oxygen evolution dictate the resulting carbon skeleton structure and thus its ability to graphitize [25]. Unlike oxygen, nitrogen and sulfur are thermally more stable in the carbon scaffold, and hence they affect the carbon structure in a very different way than oxygen [26,27]. They play a limited role in the initial low-temperature carbonization. Intense thermal desorption of sulfur is observed above ~800 °C, resulting in a significant increase in microporosity, while nitrogen is thermally stable above 1000 °C [27]. Nevertheless, all such mechanistic studies are out of the scope of this report. Here, we focused on experimental studies regarding the effect of pyrolysis under chlorine on the porosity and elemental composition of the obtained chars.

Table 2 shows the carbon materials yield of the high-temperature treatments performed under various atmospheres. A few interesting trends were observed. Generally, pyrolysis under chlorine results in a lower yield than pyrolysis under helium. In particular, concentrated chlorine conditions resulted in a much lower yield in comparison with diluted Cl_2_ or an inert atmosphere. As expected, ammonia annealing caused additional mass loss, which was correlated with increases in S_BET_ values, as discussed below. In most cases, the chars obtained under chlorine were in the form of loose powders (as the initial polymers). However, some of them underwent sintering (coking) into one piece of hard char, which resulted in low S_BET_ values (e.g., the N-doped sample under concentrated Cl_2_). Importantly, this phenomenon (sintering/coking) did not occur under a He atmosphere.

Table 3 presents the S_BET_ values of the carbons obtained via pyrolysis (and additional NH_3_ annealing) under various conditions. The N-doped polymer yielded carbons of very low S_BET_ values under He or concentrated Cl_2_ (90%). Its porosity developed well only under diluted Cl_2_ (10%). This observation shows that an optimal Cl_2_ concentration exists (around low concentration values) for the N-doped sample, under which a highly microporous carbon can be obtained. This is a very distinctive property of this material compared with O- and S-doped polymers. We discussed the peculiarity of the N-doped resin in detail elsewhere [21]. The O-doped polymer yielded carbon of much higher S_BET_ values under a Cl_2_ atmosphere than under an inert atmosphere (1075 vs. 640 m^2^g^−1^). Moreover, the high S_BET_ values were independent of chlorine concentration. This was a significant difference between the O-doped and N-doped polymers. Similar to the O-doped polymer, the S-doped resin yielded carbons with similar S_BET_ values regardless of the chlorine concentration. However, in this case, the S_BET_ values were significantly lower and quite similar regardless of the carbonization atmosphere (inert vs. oxidative). Similarities in S_BET_ values regardless of the pyrolysis atmosphere (He vs. Cl_2_) and chlorine concentration is a distinctive property of the S-doped material.

In all cases, additional NH_3_ annealing caused a significant increase in S_BET_ values. Due to the very low S_BET_ values of the N-doped polymer after pyrolysis under He and concentrated Cl_2_, we did not subject it to additional NH_3_ annealing. The most significant increase was observed for the S-doped and N-doped samples pyrolyzed under helium (for the S-doped sample) and diluted chlorine (for the N-doped sample). Pyrolysis under He yielded samples of relatively low S_BET_ values, which in the case of the O-doped polymer were only moderately enhanced by NH_3_ annealing. By contrast, chlorine pyrolysis allowed the synthesis of carbons with S_BET_ values of up to 1415 m^2^g^−1^ after additional ammonia treatment. Large increases in S_BET_ values after NH_3_ treatments were accompanied by significant mass losses of 30–40%, and hence a lower carbon yield (Table 2).

Table 4 shows the micropore (V_mi_) and total pore (V_0.99_) volumes for the obtained carbonaceous materials. Generally, pyrolysis under a chlorine atmosphere yields microporous materials regardless of the chlorine concentration. The porosity of O- and S-doped samples obtained via chlorine pyrolysis can be considered an exclusively microporous structure. In fact, O- and S-doped samples show similar porous structures regardless of the pyrolysis conditions. However, the N-doped sample behaves a bit differently. A microporous carbon is only obtained in the conditions of diluted chlorine exclusively. If one compares the microporosity and S_BET_ values of the chars obtained under Cl_2_ and He, then more significant differences are observed for polymers/carbons containing elements of higher electronegativity (C = 2.55 vs. S = 2.58, N = 3.04, and O = 3.44). It could be hypothesized that the more electronegative N and O increase the cationic character of the neighboring carbons, making them more vulnerable to chlorine attack. In addition, unlike furan or thiophene rings, the pyrrolic ring possesses protonated nitrogen (N‒H functionality), which can be especially prone to reaction with chlorine. However, under diluted chlorine, the chlorination process is less intense and yields N-doped carbons of developed porosity. Due to the presence of protonated nitrogen, the pyrrole ring forms hydrogen bonds, and this ability differentiates it from furan and thiophene. Additionally, 2-formylpyrrole is the most reactive of all the heteroaldehydes used in this study, as the sol-to-gel transition it causes occurs almost instantaneously. We previously reported some striking differences between carbons derived from pyrrole-based aldehydes vs. furan- and thiophene-based aldehydes [21,27]. N-doped carbons derived from pyrrole-based aldehydes possess unusual narrow microporosity (called ultramicropores) inaccessible by N_2_ molecules at liquid nitrogen temperatures [21].

Table 5 presents an elemental analysis of the obtained carbonaceous materials. One must bear in mind that combustion CHNS analysis based on the classical Pregl–Dumas method does not assess the oxygen and chlorine contents in the analyzed samples, and hence the C, H, N, and S contents do not add up to the total content of all elements. The decreased amount of carbon in samples obtained under Cl_2_ in comparison with samples obtained under He is caused by the chemisorption of this heavy element on the carbon surface. Heavy chlorine atoms contribute significantly to the overall sample mass. After ammonia annealing, chlorine is removed, and hence the relative carbon content grows significantly, exceeding the values observed for samples pyrolyzed under helium. Sulfur and nitrogen chemisorb strongly on carbonaceous materials. In fact, they can replace carbon atoms in the sp^2^ hybridized honeycomb lattice yielding substitutionally doped graphenic carbon materials. An appreciable amount of sulfur (~2 wt.%) can be retained in the carbon structure even after heating to 1200 °C [27,28]. However, sulfur can be completely removed (and evolved as H_2_S) by heat treatment in hydrogen at over 700 °C. Similarly, ammonia annealing desulfurizes carbonaceous materials (3 −S_(chemisorbed)_ + 2NH_3_ → 3H_2_S + N_2_). High-temperature treatment under ammonia is a very common method for simultaneously introducing nitrogen functionalities to porous carbon surfaces and increasing their basicity [29]. For this reason, we observed efficient desulfurization of the S-doped sample under additional NH_3_ annealing accompanied by the introduction of nitrogen. The N-doped samples exhibited a similar nitrogen content regardless of the pyrolysis atmosphere. This shows that chlorine does not remove nitrogen from the carbonaceous material. For the O-doped samples, the additional ammonia annealing introduced up to ~2 wt.% of nitrogen. Very interesting CHNS analysis results were obtained for the S-doped sample. Pyrolysis under concentrated chlorine resulted in a somewhat diminished amount of sulfur in comparison with pyrolysis under He or diluted Cl_2_. In addition, in all cases, ammonia annealing caused a drastic reduction in S content and a significant increase in N content (Table 5).

Finally, it must also be noted that the presence of chlorine in the samples pyrolyzed under a Cl_2_ atmosphere (i.e., without the ammonia annealing step) may in fact be advantageous. More and more reports indicate that chlorine-doped and chlorine-containing multi-doped porous carbons have a number of advanced applications [30,31,32,33]. Such materials have already been shown to be efficient electrocatalysts for nitrogen and oxygen reduction and, more generally, efficient functional materials for energy storage/conversion devices. Moreover, we recently demonstrated that direct pyrolysis of N-doped polymers with the addition of an iron source yields active Fe–N–C electrocatalysts with S_BET_ values of up to ~1500 m^2^g^−1^ [34].

## 4. Conclusions

Pyrolysis of porous organic synthetic polymers under an oxidative chlorine atmosphere yielded microporous carbonaceous materials with reasonably high yields, which decreased inversely with chlorine concentration. Generally, the pyrolysis yield under chlorine is lower than that under an inert atmosphere. Nevertheless, carbonization under Cl_2_ produces carbons of higher specific surface area values and more enhanced microporosity compared with pyrolysis in inert conditions. The N and S heteroatom content in the final chars was not strongly reduced by chlorine pyrolysis compared with pyrolysis under helium. However, some chlorine can be retained in the porous carbons upon pyrolysis at 800 °C, but its removal can be efficiently achieved via subsequent ammonia annealing. NH_3_ treatment of the chlorine carbonized samples significantly increased the relative carbon and nitrogen contents. It also efficiently desulfurized the S-doped carbons. In conclusion, direct pyrolysis of organic thermosetting resins yields heteroatom-doped microporous carbons with S_BET_ values of up to 1100 m^2^g^−1^, which can increase up to 1415 m^2^g^−1^ upon ammonia annealing. Consequently, direct carbonization under chlorine can be considered as an innovative approach to synthesize chlorine-doped and multi-doped carbonaceous materials.

## Figures and Tables

**Figure 1 molecules-26-03656-f001:**
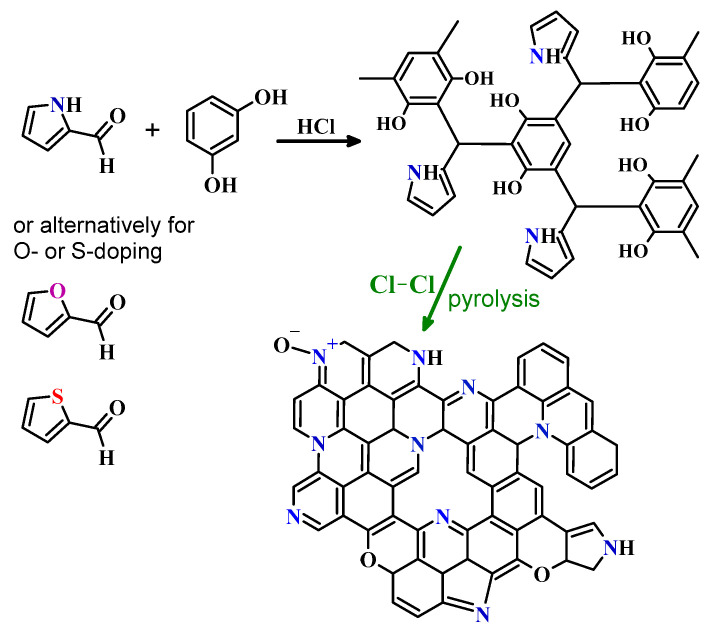
Schematic representation of the polycondensation reaction, the cross-linked polymer structure, the utilized heteroaromatic aldehydes, and pyrolysis under chlorine.

**Table 1 molecules-26-03656-t001:** Elemental composition (wt.%) of the studied organic polymers.

Organic Polymer	C	H	N	S	O
N-doped	63.14	4.02	9.66	-	23.18
O-doped	65.63	3.84	-	-	30.53
S-doped	60.61	3.53	-	19.74	16.12

**Table 2 molecules-26-03656-t002:** Total yields of pyrolysis and additional ammonia annealing of the performed treatments.

Sample	Pyrolysis/Additional Ammonia Annealing Yield (%)					
	He	+NH_3_	Cl_2_ (10%)	+NH_3_	Cl_2_ (90%)	+NH_3_
N-doped	55.2	n.d.	53.3	32.0	28.6	n.d.
O-doped	51.0	45.7	38.8	30.1	32.1	22.1
S-doped	52.2	36.5	45.8	32.2	38.2	27.0

**Table 3 molecules-26-03656-t003:** Influence of pyrolysis conditions and additional ammonia annealing on the S_BET_ values (m^2^g^−1^) of the obtained carbon materials.

Sample	BET Specific Surface Area					
	He	+NH_3_	Cl_2_ (10%)	+NH_3_	Cl_2_ (90%)	+NH_3_
N-doped	65	n.d.	765	1280	50	n.d.
O-doped	640	790	1075	1415	1035	1300
S-doped	665	1270	770	1205	755	1170

**Table 4 molecules-26-03656-t004:** Influence of pyrolysis conditions and additional ammonia annealing on micropore (V_mi_) and total pore (V_0.99_) volume values (cm^3^g^−1^) of the obtained carbon materials.

Sample	Pyrolysis and Additional Treatment Conditions																	
	He			+NH_3_			Cl_2_ (10%)			+NH_3_			Cl_2_ (90%)			+NH_3_		
	**V_mi_**	**V_0.99_**	**%V_mi_**	**V_mi_**	**V_0.99_**	**%V_mi_**	**V_mi_**	**V_0.99_**	**%V_mi_**	**V_mi_**	**V_0.99_**	**%V_mi_**	**V_mi_**	**V_0.99_**	**%V_mi_**	**V_mi_**	**V_0.99_**	**%V_mi_**
N-doped	0.03	0.09	33	-	-	-	0.29	0.32	91	0.51	0.58	88	0.02	0.02	100	-	-	-
O-doped	0.24	0.26	92	0.30	0.31	97	0.42	0.43	98	0.56	0.57	98	0.40	0.43	93	0.52	0.58	90
S-doped	0.25	0.27	92	0.50	0.51	98	0.30	0.33	91	0.48	0.49	98	0.30	0.31	97	0.47	0.48	98

**Table 5 molecules-26-03656-t005:** Influence of pyrolysis conditions and additional ammonia annealing on the elemental composition (wt.%) of the obtained carbon materials.

Sample	Pyrolysis and Additional Treatment Conditions																							
	He				+NH_3_				Cl_2_ (10%)				+NH_3_				Cl_2_ (90%)				+NH_3_			
	C	H	N	S	C	H	N	S	C	H	N	S	C	H	N	S	C	H	N	S	C	H	N	S
N-doped	82.75	1.23	8.17	-	-	-	-	-	67.10	1.40	7.67	-	84.84	1.13	8.13	-	71.95	0.76	7.54	-	-	-	-	-
O-doped	94.78	0.80	-	-	94.80	0.78	2.10	-	81.93	0.37	-	-	95.28	0.52	2.14	-	84.37	0.37	-	-	93.90	0.43	1.10	-
S-doped	86.70	0.60	-	9.45	88.82	0.98	6.85	0.53	76.00	0.17	-	7.60	91.12	0.85	3.90	0.83	78.15	0.27	-	4.63	91.85	0.73	4.53	0.72

## Data Availability

Data are available from the authors on request.

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
