# Peer review of "Pyrolysis of Porous Organic Polymers under a Chlorine Atmosphere to Produce Heteroatom-Doped Microporous Carbons"

_molecules, 2021, doi:10.3390/molecules26123656_

Round 1

Reviewer 1 Report

The current manuscript by Kiciński, Dyjak, and Gratzke presents a short investigation of the porosity of carbon-based materials upon pyrolysis. The study presents a pathway for improvement of the specific surface area of the carbon-based materials upon doping them with different gases upon pyrolysis. The study itself is short and lacks a description of the materials used (dimensions, porosities, densities, etc.). It is also scarce on experimental results for the characterization of the materials, e.g. changes in the porosity distributions, SEM/TEM characterization of micro and nanopores, mechanical integrity/strength, etc. This hinders the readers' interest in further development of the topic. From this point of view, I find the publication of the article a bit premature, and I recommend a major revision. Minor technical errors along the text should also be corrected. For instance, there is a common misuse of super- and sub-scripts:
“ml min−1” should be in superscript “ml.min−1
“m2g−1” should be “m2.g-1,
“SBET” should read “SBET
“NH3” should be “NH3
 “H2S” should be “H2S”

Author Response

Detailed responses to the Reviewers’ comments:

Reviewer 1

The study itself is short and lacks a description of the materials used (dimensions, porosities, densities, etc.). It is also scarce on experimental results for the characterization of the materials, e.g. changes in the porosity distributions, SEM/TEM characterization of micro and nanopores, mechanical integrity/strength, etc. This hinders the readers' interest in further development of the topic. From this point of view, I find the publication of the article a bit premature, and I recommend a major revision.

Minor technical errors along the text should also be corrected. For instance, there is a common misuse of super- and sub-scripts:
“ml min−1” should be in superscript “ml.min−1
“m2g−1” should be “m2.g-1,
“SBET” should read “SBET
“NH3” should be “NH3
 “H2S” should be “H2S”

Response:

Reviewer 1 noticed that the report is short and lacks extensive characterization of the utilized polymers and the obtained carbon materials. However, please note that the very aim of this study was only to investigate the pyrolysis under chlorine and its impact on porosity and chemical composition. We kept the paper very short for a good reason – we indeed planed carefully to keep it brief. We did highlight the very narrow focus of this study (as we do say in the paper: “The undertaken research has two very well-focused aims, i.e., to study the influence of carbonization under chlorine on the porosity and elemental composition of porous carbonaceous materials derived from heteroatom-containing synthetic polymers and to elucidate how pyrolysis under oxidative Cl2 compares with pyrolysis under an inert gas”). Please note that the submitted manuscript belongs to a special issue: Porous Organic Polymers: Synthesis, Characterization and Applications” and for this reason the manuscript is brief and highly focused on a very specific topic. Please consider also that due to high toxicity and reactivity of chlorine its utilization at high temperature is challenging and hence herein we have focused on these issues more than on excessive characterization of the final carbonaceous products. We would like to apologies for the common misuse of super- and sub-scripts in the manuscript send for peer review – this was an error during generating PDF and we did correct diligently all the mistakes. We also added some new references to frame this report in a broader context and show a bigger picture. Finally, we have replaced Figure 1 with one of much better quality. We would like to highlight that we corrected all the editorial mistakes and errors very diligently and now the revised manuscript looks much better.

Again, please accept our sincere apologies for the first, unpolished version of the manuscript. We do appreciate the time and effort you have invested to read it in the previous form with many editorial shortcomings.

Reviewer 2 Report

The authors claim to have developed a process by which it is possible to introduce porosity in carbon materials. This method is based on pyrolysis in a chlorine based environment and annealing in ammonia at elevated temperatures. I appreciate the authors for simple writing that is easy to understand. The development of new material development process has applications in separation applications and is important.

My queries are,

  1. Is there any error bar for the measurements of surface area, internal volume. I would like to know the degree of repeatability and reliability of this study
  2. In the measurement of BET surface area, internal volume and elemetal composition, there are some gaps in the data presented. Why is it so.
  3. While the 10% CL2 atmosphere seem to yield better surface area compared to 90% Cl2, the reasons are not discussed clearly. A brief discussion on mechanics of the reaction (maintext or suplemental info) with appropriate reference would help readers understand the underlying mechanics. Specific elucidation of why 10% Cl2 yields better surface area and how doping affects it needs to be elucidated.

Author Response

Reviewer 2

  1. Is there any error bar for the measurements of surface area, internal volume. I would like to know the degree of repeatability and reliability of this study

Response:

This is indeed a very interesting and very important question as the data of specific surface area (SSA) determined by BET method is very often reported incorrectly even in very competitive journals. To answer this question, we cite some critical recommendations from two very important papers:

  • Thommes, et. al., Physisorption of gases, with special reference to the evaluation of surface area and pore size distribution (IUPAC Technical Report). Pure Appl. Chem. 87 (2015) 1051–1069. https://doi.org/10.1515/pac-2014-1117
  • Badalyan and P. Pendleton, Analysis of Uncertainties in Manometric Gas-Adsorption Measurements. I: Propagation of Uncertainties in BET Analyses, Langmuir 19 (2003) 7919-7928. https://doi.org/10.1021/la020985t

In fact, we included both articles in the list of references. Generally, the calculated value of the BET area is dependent on the adsorptive and operational temperature and the procedure used to locate the pressure range in applying the BET equation. For microporous materials assessment of specific surface area by BET method is especially challenging since there can be an appreciable overlap of monolayer and multilayer adsorption. It is important to note than in some particular cases the BET method is even not applicable to evaluate SSA. The range of linearity of the BET plot is always restricted to a very limited part of the obtained isotherm. Nitrogen (at its boiling temperature, 77 K) is traditionally the adsorptive commonly used to determine the SBET. This is partly because liquid N2 is readily available. However, due to its quadrupole moment, the orientation of a nitrogen molecule is dependent on the surface chemistry of the adsorbent. This may lead to high uncertainty in the value of molecular cross-sectional area of N2 – possibly 20% for some surfaces. The BET method can be applied to many Type II and Type IV isotherms, but extreme caution is needed in the presence of micropores (i.e., with Type I isotherms and combinations of Types I and II or Types I and IV isotherms – and this is indeed our case). In this instance it may be impossible to separate the processes of monolayer/multilayer adsorption and micropore filling. With microporous adsorbents, the linear range of the BET plot may be indeed very difficult to locate. As recommended by IUPAC, the BET-area derived from a Type I isotherm should not be treated as a realistic probe accessible surface area. It rather represents an apparent surface area. Accuracy of SBET can be also assessed based on a reference material provided with the utilized apparatus. For instance, we regularly monitor the quality of our results utilizing two reference materials: one which exhibits a specification of 755.06 m2g-1 ± 68.70 m2g-1 (multi-point SBET) and another which exhibits a specification of 97.82 m2g-1 ± 7.53 m2g-1 and this yields overall accuracy of ca. 8−9 %. In consequence, we can expect such level of accuracy, but that depends also on the specific characteristics of the measured samples. For instance, the lower the SBET of a sample is and the lower the mass of a sample used for analysis then the lower the overall accuracy. It is generally accepted that the error in the measurement of the SSA by nitrogen at its boiling point is around 5-10 % and the lowest surface areas that can be actually measured with reliable precision is of ~10 m2. For samples with lower SSAs Kr should be used as the adsorptive instead of N2. But generally, for samples with such a low SSA a bigger mass should be used for measurement. Please note that we added appropriate comment and literature dealing with this issue in the revised manuscript.

  1. In the measurement of BET surface area, internal volume and elemental composition, there are some gaps in the data presented. Why is it so.

Response:

Due to very low SBET value after pyrolysis under He and under concentrated Cl2 for the N-doped polymer we decided not to subject it for additional NH3 annealing. In case of elemental CHNS compositions, the gaps are also caused by absence of certain heteroatoms in the raw organic polymer, for instance the O-doped polymer does not contain any nitrogen and sulfur, but nitrogen appears in all the NH3 treated samples.

  1. While the 10% Cl2 atmosphere seem to yield better surface area compared to 90% Cl2, the reasons are not discussed clearly. A brief discussion on mechanics of the reaction (maintext or suplemental info) with appropriate reference would help readers understand the underlying mechanics. Specific elucidation of why 10% Cl2 yields better surface area and how doping affects it needs to be elucidated.

Response:

Indeed, maybe we were not clear enough at this point, but please note that  we did elucidate in the manuscript that:

The N-doped polymer yields carbons of very low SBET values under He or concentrated Cl2 (90%). Its porosity develops well only under diluted Cl2 (10%). This observation shows that an optimal Cl2 concentration exists (around low concentration values) for the N-doped sample, under which a highly microporous carbon can be obtained. It is a very distinctive property of this material when compared to O- and S-doped polymers. We have discussed the peculiarity of the N-doped resin in detail elsewhere [21]. The O-doped polymer yields carbon of much higher SBET values under a Cl2 atmosphere than under an inert atmosphere (1075 vs 640 m2g−1). Moreover, the high SBET values are independent of chlorine concentration. This is a significant difference between the O-doped and N-doped polymers. Similarly to the O-doped polymer, the S-doped resin yields carbons of relatively similar SBET regardless of the chlorine concentration. Similarities in SBET values regardless of pyrolysis atmosphere (He vs Cl2) and chlorine concentration is a very distinctive property of the S-doped material”.

Also, we do elucidate that:

It could be hypothesized that the more electronegative N and O increase the cationic character of the neighboring carbons, making them more vulnerable to chlorine attack. Also, unlike furan or thiophene rings, the pyrrolic ring possesses protonated nitrogen (N‒H functionality) which can be especially prone to reaction with chlorine. However, under diluted chlorine the chlorination process is less intense and yields N-doped carbons of developed porosity.” Consequently, in our humble opinion we did the best for now trying to explain the observed phenomena and the provided description holds some new insight and scientific value.

To sum up, in our opinion we did give some strong elucidations to the discussed results.  In fact, it must be highlighted that in the case of O- and S-doped polymers direct pyrolysis under Cl2 yields very similar SBET values for concentrated and diluted chlorine atmospheres.

Round 2

Reviewer 1 Report

The authors of the study have improved the presentation of their manuscript and elucidated its relevance to the Molecules Special Issue Porous Organic Polymers: Synthesis, Characterization and Applications”, which suggests a more narrow and specialized readers than usual. As a result of the latter I recommend the manuscript for publications as it is.

Reviewer 2 Report

The authors have addressed my comments appropriately. I recommend the manuscript for publication.